# Changes in Phospholipid Composition of the Human Cerebellum and Motor Cortex during Normal Ageing

**DOI:** 10.3390/nu14122495

**Published:** 2022-06-16

**Authors:** Sarah E. Hancock, Michael G. Friedrich, Todd W. Mitchell, Roger J. W. Truscott, Paul L. Else

**Affiliations:** 1Victor Chang Cardiac Research Institute, Darlinghurst, NSW 2010, Australia; s.hancock@victorchang.edu.au; 2Illawarra Health and Medical Research Institute, University of Wollongong, Wollongong, NSW 2522, Australia; michaelf@uow.edu.au (M.G.F.); toddm@uow.edu.au (T.W.M.); rjwt@uow.edu.au (R.J.W.T.); 3School of Medical, Indigenous and Health Sciences, University of Wollongong, Wollongong, NSW 2522, Australia

**Keywords:** mass spectrometry, lipidomics, mitochondria, microsome, neurodegeneration, docosahexaenoic acid, arachidonic acid, adrenic acid, prefrontal cortex, hippocampus, entorhinal cortex

## Abstract

(1) Background: Changes in phospholipid (phosphatidylcholine, phosphatidylethanolamine and phosphatidylserine, i.e., PC, PE and PS) composition with age in the mitochondrial and microsomal membranes of the human cerebellum and motor cortex were examined and compared to previous analyses of the prefrontal cortex, hippocampus and entorhinal cortex. (2) Methods: Nano-electrospray ionization on a hybrid triple quadrupole–linear ion trap mass spectrometer was used to analyse the brain regions of subjects aged 18–104 years. (3) Results: With age, the cerebellum showed many changes in the major phospholipids (>10% of the phospholipid class). In both membrane types, these included increases in PE 18:0_22:6 and PS 18:0_22:6, decreases in PE 18:0_20:4 and PS 18:0_18:1 and an increase in PC 16:0_16:0 (microsomal membrane only). In addition, twenty-one minor phospholipids also changed. In the motor cortex, only ten minor phospholipids changed with age. With age, the acyl composition of the membranes in the cerebellum increased in docosahexaenoic acid (22:6) and decreased in the arachidonic (20:4) and adrenic (22:4) acids. A comparison of phospholipid changes in the cerebellum, motor cortex and other brain areas is provided. (4) Conclusions: The cerebellum is exceptional in the large number of major phospholipids that undergo changes (with consequential changes in acyl composition) with age, whereas the motor cortex is highly resistant to change.

## 1. Introduction

The brain is the organ we tend to give the most credit to for our accomplishments as a species. Apart from coordinating our crucial physiological functions, the human brain appears to be unique in the extent to which it has allowed us to produce social constructs and imagine subjective entities well outside the objective realties of our natural world. About 50% of the dry weight of the brain is lipid, and almost half of that is phospholipid associated with membranes [1]. In the past, brain lipids have been analysed mainly using thin-layer and gas chromatography. These techniques have provided us with information about the relative abundance of phospholipid classes and details about the gross fatty acid composition of each class. However, recent advances in mass spectrometry (i.e., lipidomics) have allowed for the analysis of the phospholipid composition at an unrivalled level, i.e., the specific phospholipid type and fatty acid combinations. The lipid composition of the brain is known to change with age; however, much of this work has tended to concentrate on changes in the lipid composition associated with age-related pathologies (e.g., Alzheimer’s). Far fewer studies have used lipidomics to analyse changes in the brain lipid composition during normal ageing and how this might change in different areas of the brain.

The phospholipid composition of the brain, at a gross level, is similar to that of other tissues (see Table 4 in [2]), with the molecular “species” phosphatidylcholine (PC) and phosphatidylethanolamine (PE) making up the majority of the phospholipids (~75%), followed by phosphatidylserine (PS) at ~20%. This study examines these three major phospholipid classes (PC, PE and PS) that constitute the majority of brain phospholipids, rather than concentrating on the phospholipid classes in lower abundances such as sphingomyelin, phosphatidic acid, phosphatidylinositol, cardiolipin and phosphatidylglycerol. One major difference between the brain and other tissues is that the brain has a higher level of omega-3 fatty acids associated with its phospholipids. However, although it is viewed as a highly polyunsaturated organ, the brain has an overall lower percentage of polyunsaturated fatty acids (PUFA) compared to other major organs [2].

The original study from which this study’s data are drawn involved an examination of the changes in the phospholipid composition of five different human brain regions during natural ageing, with subjects aged from 18 to 104 years of age. To further delineate where the differences in membrane phospholipid composition may occur during ageing, brain tissue from the different regions was fractionated into a mitochondrial-enriched fraction and a microsomal membrane fraction (predominately the Golgi, endoplasmic reticulum, vesicles and plasma membrane) using differential centrifugation. Results for the prefrontal cortex [3], hippocampus [4] and entorhinal cortex [5] have previously been published. The present study reports on the results of the two remaining human brain areas studied, namely the cerebellum and the motor cortex. In addition, it provides a unique comparison of the changes in the phospholipid composition that occur with normal ageing for the mitochondrial and microsomal membranes across all five areas of the human brain.

## 2. Materials and Methods

### 2.1. Materials

All solvents (e.g., methanol and chloroform) were of HPLC grade or higher and were purchased from VWR International (Brisbane, Australia). The ammonium acetate (analytical grade) was obtained from Crown Scientific (Minto, Australia). Zirconium oxide beads (1.4 mm) were purchased from Sapphire Biosciences (Sydney, Australia). The sucrose, tris, dithiothreitol (DTT) and ethylenediaminetetraacetic acid (EDTA) were purchased from Astral Scientific (Sydney, Australia). Pierce BCA Protein Assay Kits were obtained from Thermo Fisher Scientific (Melbourne, Australia). The protease inhibitor cocktail and butylated hydroxytoluene (BHT) were purchased from Sigma Aldrich (Sydney, Australia). Internal glycerophospholipid standards were purchased from Avanti Polar Lipids through Auspep (Melbourne, Australia). The standard used consisted of 20 µM each of PC 19:0_19:0, PE 17:0_17:0 and PS 17:0_17:0, and 10 µM each of lyso-PC 17:0 and lyso-PE 14:0 in chloroform: methanol (2:1, *v*/*v*).

### 2.2. Brain Tissue

Frozen neurologically normal post-mortem brain specimens from five brain regions (i.e., the prefrontal cortex, hippocampus, entorhinal cortex, cerebellum and motor cortex, with the motor cortex and cerebellum specifically examined in this study) were obtained from the New South Wales Tissue Resource Centre at the University of Sydney (PID0215). The experiments were conducted according to approvals provided by the Human Research Ethics Committee of the University of Wollongong (HE11/267, 1 June 2012). Specific details (i.e., age, gender, post-mortem interval and cause of death) of the 36 donor subjects (24 male and 12 female) from which the brain samples were obtained have been previously published and are available in Table 1 of [3]. The average age of the donors was 59 years, with the females being slightly older than the males (71.4 ± 7 versus 52.7 ± 3.9 years). Although the brains were obtained from subjects aged from 18 to 104 years, all comparative analyses were based on the changes (using linear regression) at 20 years compared to those at 100 years—an 80-year adult age span. Brains from the donors over the age of 60 years underwent a modified Braak staging screen (by the New South Wales Tissue Resource Centre). Stages I and II (of the six progressive stages) were observed in some elderly donors, but the donors were deemed neurologically normal for their age. The samples were shipped on dry ice and stored at −80 °C until required, whereupon they were pulverized using dry ice prior to being accurately weighed for subcellular fractionation.

### 2.3. Subcellular Fractionation

Pulverized brain tissue (~100 mg) was homogenized using a FastPrep instrument (MP Biomedicals, Sydney, NSW, Australia) with zirconium oxide beads in an ice-cold buffer (250 mM sucrose, 20 mM Tris, 2 mM of DTT, 2 mM of EDTA and a complete protease inhibitor at pH 7.4). The homogenate was centrifuged for 10 min at 1000× *g* to remove nuclei and large cellular debris. The supernatant was then centrifuged for 35 min at 10,000× *g* to produce a pellet enriched in mitochondria, which was collected. The supernatant was retained and centrifuged for a further 40 min at 100,000× *g* to produce a microsomal pellet and cytosolic supernatant. Both the mitochondrial and microsomal pellets were resuspended in milliQ H_2_O and the total protein content was determined (BCA Protein Assay Kit, Thermo Fisher Scientific, Scoresby, VIC, Australia). A linear regression analysis for changes in the total protein with age in the whole tissue homogenate, mitochondrial fraction and microsomal fraction did not show any age-related changes.

### 2.4. Lipid Extraction

Aliquots (75 µg of total protein) of the mitochondrial and microsomal fractions from the cerebellum and motor cortex were added to chloroform:methanol (2 mL, 2:1 *v*/*v* with 0.01% BHT) together with 50 µL of the internal standard mixture (described in Section 2.1 Materials). The lipids were extracted using a modified Folch method as previously described [6]. The extracted lipids were reconstituted in chloroform:methanol (1 mL, 1:2 *v*/*v* with 0.01% BHT) and stored at −20 °C until analysed.

### 2.5. Mass Spectrometry and Lipid Analysis

A nano-electrospray ionization (nanoESI) mass spectrometry analysis of lipid extracts was conducted using a hybrid triple quadrupole–linear ion trap mass spectrometer (QTRAP^®^ 5500 AB Sciex, MA, USA) equipped with an automated chip-based nanoESI system (TriVersa Nanomate™, Advion Biosciences, Ithaca, NY, USA). To assess the total phospholipids, the samples were diluted to approximately 10 µM, then spiked with 5 mM of ammonium acetate before being loaded onto a 96-well plate and centrifuged for 10 min at 2200× *g* prior to direct infusion. The spray parameters were set at a gas pressure of 0.4 psi and voltages of 1.2 and 1.1 kV for the positive and negative ion modes, respectively, for all acquisitions. The phospholipid data were acquired by targeted ion scans using multi-channel acquisition (setting available in Table S2 [3]).

Using internal standards, Lipidview™ (version 1.2, AB Sciex, MA, USA) was used to quantify the phospholipid species in the positive ion head group scans. The software settings used included a mass tolerance of 0.5 Da, with a minimum intensity of 0.1% and a minimum signal-to-noise ratio of 10. Any phospholipid species comprising less than 0.5% of each phospholipid class across the cohort was removed from the analysis. Each phospholipid quantified was assessed for possible isobaric molecular species using the lipid catalogue tool within Lipidview™. Confirmatory ions from the negative ion fatty acid precursor scans were used to identify the amount of each molecular species present in each sample. Isobaric phospholipid species that contained either an odd-chain or ether-linked fatty acid were unable to be separated using this method and were reported as being either species. Due to this, no correction factor was able to be applied to the PE ether phospholipid species. Phospholipid molecular species that appeared in less than 75% of samples were removed prior to the statistical analysis. The reported phospholipids used the nomenclature outlined in [7] and were reported as normalized values relative to the total phospholipid detected in each phospholipid class to reduce the variation between the samples.

### 2.6. Statistical Analysis

Most statistical analyses were performed using SPSS Statistics (version 19, IBM Corp., New York, NY, USA). The Wilcoxon signed-rank test was used for all comparisons of phospholipid classes between the mitochondrial and microsomal fractions. To examine the relationship between age and phospholipids, a linear regression was used with the significance set at a level of *p* < 0.05 (sex was included as a second independent variable). The normality of the dependent variable was assessed by examining the histograms of the residuals, and the non-normal data were transformed where required. Outliers were identified as being three standard deviations from the mean, and were either transformed or removed from the analysis if they violated the normality assumptions. Corrections for multiple comparisons were not applied due to the exploratory nature of this study. Heat maps were created in R (v. 3.1.1) using the “gplots” package. Hierarchical clustering was used to look for any similarities in age-related phospholipid changes within the two membrane fractions examined across the five brain regions (the cerebellum, motor cortex, prefrontal cortex, hippocampus and entorhinal cortex). The normalised slope of the phospholipids, transformed for linear regression, was calculated from the back-transformed amounts present at 20 and 100 years of age to allow for comparison with the non-transformed regressions.

## 3. Results

### 3.1. Changes in Protein Concentration and Proportion of Phospholipid Classes in the Microsomal and Mitochondrial Membrane Fractions of the Human Cerebellum and Motor Cortex

The cerebellum, unlike the motor cortex, showed a decrease in its protein concentration with age (between 20 and 100 years) in both the mitochondrial (by 54%; *p* < 0.007) and microsomal (by 38%; *p* < 0.011) membrane fractions, whereas the motor cortex showed no significant change during ageing in either membrane fraction. The microsomal fraction of the cerebellum had a slightly higher proportion of PC (48 vs. 43%; *p* < 0.001) and a lower proportion of PE (33 vs. 36%; *p* < 0.001) and PS (19 vs. 22%; *p* < 0.001) than the mitochondrial membrane. In the motor cortex, the only difference was a slightly higher proportion of PS in the microsomal versus mitochondrial membrane fractions (18 vs. 16%; *p* < 0.01). When comparing the cerebellum and motor cortex, both membrane fractions tended to have a similar level of PS, at approximately 19%, but the cerebellum had a lower proportion of PC (average of 45.5 vs. 55.5%; *p* < 0.001) and a higher overall proportion of PE (34.5 vs. 27%; *p* < 0.001) compared to the motor cortex.

### 3.2. Abundant Phospholipid Molecules of the Human Cerebellum and Motor Cortex

The most abundant phospholipid molecule present in both the cerebellum and motor cortex (in both membrane fractions) was PC 16:0_18:1 (at 39% of the PC in the cerebellum and 42% of the PC in the motor cortex). This phospholipid did not change significantly with age in either membrane fraction from the two different brain areas (and therefore is not shown in Table 1). The second most abundant phospholipid species in both the cerebellum and motor cortex was PE 18:0_22:6, at 29% of the PE in the microsomal and mitochondrial membrane fractions of the cerebellum and between 35 and 37% of the PE in the same two membrane fractions of the motor cortex. PE 18:0_22:6 did change significantly with age (by 20–21%) in both membrane fractions of the cerebellum, but not in the membrane fractions of the motor cortex (see Table 1).

### 3.3. Changes in Phosphatidylcholine Molecules of the Human Cerebellum with Age

Table 1 summarises all age-related changes, including PC in both the microsomal- and mitochondrial-enriched fractions of the cerebellum and motor cortex, between the ages of 20 and 100 years. In the cerebellum, the most abundant PC molecule that increased with age was PC 16:0_16:0 (30%), representing 14% of all PC; however, this change was restricted to the microsomal membrane fraction only (Table 1 and Figure 1). The lower-abundance phospholipid, PC 16:0_18:2, at 1.2% of the microsomal and 1.1% of the mitochondrial PC, increased with age in both membrane fractions (54% and 39% in the microsomal and mitochondrial membrane fractions, respectively), whereas in the mitochondrial-enriched fraction, the moderately abundant phospholipid PC 16:0_22:6, at 4% of PC, also increased with age (20%).

The PC molecules that decreased with age in both the microsomal and mitochondrial membrane fractions are shown in Table 1. The PC molecules that decreased in both membrane types included PC 16:0_20:1 (23 and 27%, respectively) and PC 16:0_22:4 (22 and 25%, respectively). PC 16:0_22:4 is a moderately abundant PC molecule constituting 2.5% of the microsomal and 2.9% of the mitochondrial PC, whereas PC 16:0_20:1 is a less abundant PC, at 1.5% of PC in both membrane fractions. Three other PC molecules also decreased with age, but only in the microsomal membrane fraction of the cerebellum. These included lyso PC 18:1 (10%), PC 18:0_18:1 (18%) and PC 18:1_22:6 (59%). PC 18:0_18:1 is an abundant PC molecule constituting 7.6% of all PC molecules in the cerebellum, whereas lyso PC 18:1 and PC 18:0_22:6 are both lower-abundance molecules, at 0.9% and 1.1% of the PC in the cerebellum, respectively (see Table 1).

### 3.4. Changes in Phosphatidylethanolamine and Phosphatidylserine Molecules of the Human Cerebellum with Age

With age, the microsomal and mitochondrial membranes of the cerebellum behaved similarly with respect to the changes in PE phospholipids, with increases in PE 16:0_22:6 (25 and 34%, respectively), PE 18:0_22:6 (20 and 21%, respectively), PE 18:1_22:6 (26 and 33%, respectively) and PE O-16:0_22:6/PE15:0_22:6 (61 and 263%, respectively), and with decreases in PE 18:0_18:1 (27 and 28%, respectively), PE 18:0_20:4 (16 and 25%, respectively) and PE 18:0_22:4 (36 and 39%, respectively) (see Table 1 for a complete list of PE and PS molecules that changed, and Figure 1 for the major PE and PS phospholipids that changed with age). The mitochondrial membrane also showed an increase in PE O-18:1_22:6 (22%) and decreases in lyso PE 22:4 (38%), PE O-16:1_22:5 (54%) and PE O-18:0_22:4/PE 17:0_22:4 (40%) with age.

Of those PE molecules that increased, PE 18:0_22:6 (at 29% of the PE molecules) was the most significant (see Figure 1), being the primary PE molecule and the second most abundant phospholipid in the microsomal and mitochondrial membranes. Among the other PE molecules that increased, PE 16:0_22:6 was a moderately abundant PE, being the third most abundant PE molecule at 6% of PE in both membrane fractions, whereas PE 18:1_22:6 was a less abundant molecule, at 2.9% of the PE in both membrane fractions. Other PE molecules that increased included PE O-16:0_22:6/PE15:0_22:6, which increased in both the microsomal and mitochondrial membrane fractions, and PE O-18:1_22:6, which increased in the mitochondrial membrane only. Both were lower-abundance molecules, at 1.7% and 2.7% of the PE, respectively, in the membranes of the cerebellum (see Table 1). By far, the most prominent PE molecule that decreased with age in the enriched microsomal and mitochondrial membrane fractions of the cerebellum was PE 18:0_20:4, being the second most abundant PE molecule at approximately 16% of PE in both membrane fractions. Of the other PE molecules that decreased with age, PE 18:0_22:4 was the next most abundant PE molecule, at approximately 3.4% of the PE in both membrane fractions of the human cerebellum, whereas all other molecules that decreased in either membrane fraction (lyso PE 22:4, PE 16:0_22:4, PE 18:0_18:1, O-16:1_22:5 and O-18:0_22:4/17:0_22:4) were of lesser abundance, ranging from 0.5 to 2.8% of the PE molecules in the membrane fractions of the human cerebellum (see Table 1).

With respect to the PS, the most prominent PS molecule to increase with age in the cerebellum was PS 18:0_22:6 (48% in the microsomal membrane and 50% in the mitochondrial membrane), being the most prevalent PS molecule at approximately 42% of the PS in both membrane fractions. The most prominent PS molecule to decrease was PS 18:0_18:1 (38% in the microsomal membrane and 30% in the mitochondrial membrane), being the second most abundant PS molecule at 29% of the PS in both membrane fractions (see Table 1 and Figure 1). The only other PS molecule to increase significantly with age was PS 18:1_20:1 (129%), a lower-abundance molecule representing 1.9% of the PS, but this increase was only significant in the microsomal membrane fraction. Apart from PS 18:0_18:1, three other PS molecules decreased with age in the cerebellum. The most abundant of these was PS 18:0_22:4 (the fourth most abundant PS molecule, at 6% of the PS), which decreased by 28% in the microsomal and 37% in the mitochondrial membrane fractions. The two other PS molecules that decreased included PS 18:0_20:1 (56%) and PS 18:0_20:2 (38%). Both these molecules decreased significantly only in the microsomal membrane fraction, and both were lower-prevalence molecules, at 2.1% and 1.1% of the PS molecules in the human cerebellum, respectively.

### 3.5. Changes in Phosphatidylcholine Molecules of the Human Motor Cortex with Age

Changes in PC molecules were far more subdued in the motor cortex compared to those found to occur in the cerebellum. In the motor cortex, PC 16:0_18:2 was the only PC molecule to increase in both membrane fractions (51 and 59% in the microsomal and mitochondrial membranes, respectively), whereas PC 14:0_16:0 also increased (15%), but only in the microsomal membrane. Likewise, only three PC molecules decreased with age: PC 18:0 _20:4 (28%) in the microsomal membrane, and PC 16:0 _20:1 (17%) and PC 16:0 _22:4 (41%) in the mitochondrial membrane fraction. These molecules are of low or moderate abundance, ranging from 1.25 to 3.6% of the PC molecules in the human motor cortex (see Table 1).

### 3.6. Changes in Phosphatidylethanolamine and Phosphatidylserine Molecules of the Human Motor Cortex with Age

Very few changes occurred with age in either the PE or PS molecules of the human motor cortex. In the case of the mitochondrial membrane, there were no changes in any PE or PS molecules with age. The few significant changes with age that did occur in the motor cortex were all associated with the microsomal membrane only. The PE molecules that increased significantly with age included PE O-16:0_22:4/15:0_22:4 (86%), PE O-16:0_22:6/15:0_22:6 (83%), PE O-18:1_22:5 (75%) and PE O-18:1_22:6 (16%). The only PE molecule that decreased with age was PE O-16:1_22:4 (45%). All PE molecules that either increased or decreased with age in the microsomal membrane of the human motor cortex were low-abundance molecules, ranging from 0.52 to 1.99% of PE. Interestingly, over the 80-year span examined in this study, no PS molecules changed significantly in their relative abundance in either membrane fraction of the motor cortex.

### 3.7. Hierarchical Clustering of Changes in Phospholipids Associated with Age in Five Areas of the Human Brain

To examine the relationships between age and changes in the phospholipid composition of all the brain areas examined in the overall study (the cerebellum, motor cortex, prefrontal cortex, hippocampus and entorhinal cortex), a regression analysis was conducted of all the significant changes occurring with age in PC, PE and PS molecules in the five brain areas. The results of this analysis were converted into a heatmap for each membrane fraction, with Figure 2 showing the results for the microsomal and mitochondrial membranes (see the Materials and Methods section for a description of the analysis). The phospholipids that underwent the largest age-related changes appear at the top of each heatmap, while those with the biggest decrease with age are at the bottom. As previously mentioned, Figure 2 shows that the phospholipid with the largest age-related increase was PS 18:0_22:6 in the cerebellum and prefrontal cortex (and also in the mitochondrial membrane of the hippocampus). The largest age-related decrease occurred in the cerebellum, in the form of PS 18:0_18:1. The cerebellum also contained the phospholipids with the next largest increase (PE 18:0_22:6) and decrease (PE 18:0_20:4) in both membrane fractions (as shown in Figure 2). By comparing the two membrane fractions, a point of difference was observed: the mitochondrial membranes in the hippocampus, cerebellum and prefrontal cortex shared PS 18:0_22:6 as the most increased phospholipid during ageing, whereas this increase was notably lacking in the microsomal membranes.

Based on the hierarchical clustering of the changes in the phospholipids associated with age (see Figure 2), the microsomal and mitochondrial membranes produced some differences in the grouping of the various brain areas. The mitochondrial membrane of the cerebellum was associated with two clustered groups: the motor and entorhinal cortices, and the hippocampus and prefrontal cortex. In the microsomal membrane, the cerebellum was associated with two separate groupings, those being the prefrontal cortex and two further clusterings, which included the hippocampus alone and the entorhinal and motor cortices as a group. Therefore, in both membrane fractions the cerebellum was distinctly different from the other brain areas. The clustering of the entorhinal and motor cortices together in both membrane fractions was largely due to the lack of age-related changes in their phospholipids. However, the microsomal membranes of the prefrontal cortex and hippocampus did not cluster together, unlike the mitochondrial membranes. This was due to a higher number of phospholipid changes identified with age within the microsomal membrane compared to mitochondrial membrane of the motor cortex. In the hierarchical clustering, the phospholipids with the greatest decrease associated with age included many PS and PE molecules, notably PS 18:0_18:1, PE 18:0_20:4 and PS 18:0_22:4 in both membrane fractions of the cerebellum and PS 18:0_22:4 in the microsomal membrane of the entorhinal cortex. In the mitochondrial membrane, the hippocampus showed a decrease in PS 18:0_22:4 and the prefrontal cortex showed a decrease in PE 18:0_20:4, emphasising the individual nature of many of these changes in terms of the brain area and the membrane type.

## 4. Discussion

The human cerebellum displayed the greatest number of changes with age in its phospholipid composition compared to the motor cortex and all other areas of the brain previously examined, including the prefrontal cortex [3], hippocampus [4] and entorhinal cortex [5]. The motor cortex was notable in displaying the fewest number of changes. Within the cerebellum, twenty-six different phospholipids were found to change with age, half of which occurred in both membrane fractions (mixed microsomal and enriched mitochondrial membranes). One of the remarkable aspects of the changes in the phospholipid composition of the cerebellum was how many of the phospholipids that changed were major lipids (shown in Figure 1). For example, in the microsomal membrane fraction, the major phospholipids that changed included PC 16:0_16:0, PE 18:0_22:6, PE 18:0_20:4, PS 18:0_22:6 and PS 18:0_18:1, which represented 14, 29, 16, 43 and 30% of their respective phospholipid classes. Similarly, in the mitochondrial-enriched membrane fraction, PE 18:0_22:6, PE 18:0_20:4, PS 18:0_22:6 and PS 18:0_18:1 changed, with these phospholipids representing 29, 16, 42 and 29% of their respective phospholipid classes. No other brain area studied has shown such large changes with age in its major phospholipids, with most brain areas possessing no, or at most only two major phospholipids (i.e., those representing greater than 10% of their class) that change with age.

In the case of the cerebellum, the changes in PE 18:0_22:6 and PS 18:0_22:6 (which changed in both membrane fractions) were particularly notable, being the most prevalent phospholipids within each of their classes (29 and 42%, respectively) and therefore having a major influence on the acyl composition of the cerebellum (discussed later). In the case of PS 18:0_22:6, our previous studies [3,4,5] have shown similar increases with age for this particular phospholipid in other brain areas, such as the prefrontal cortex [3] and the hippocampus (mitochondrial membrane only, [4]); however, the increase in PE 18:0_22:6 and the overall size of the increases and decreases in other major phospholipids within the cerebellum were unique. The cerebellum was distinct in possessing two major phospholipids, PS 18:0_18:1 (29% of the PS) and PE 18:0_20:4 (16% of the PE), that showed large decreases with age (averaging a 34% and 20% decrease in both membrane fractions, respectively). The prefrontal cortex also shared a major decrease in PE 18:0_20:4 with the cerebellum, but in the case of the prefrontal cortex, this change was only significant in the mitochondrial membrane fraction [3]—not both membrane fractions, as found in the cerebellum.

The motor cortex was remarkable for its lack of change, with those phospholipids that did change with age representing only between 0.5 and 1.7% of each phospholipid class. This lack of change was particularly conspicuous for the mitochondrial membrane, with only one minor phospholipid, PC 16:0_18:2 (at 1.3% of the PC), statistically increasing with age and two minor phospholipids, PC 16:0_20:1 (3.6% of the PC) and PC 16:0_22:4 (2.0% of the PC), decreasing with age. The overall lack of change in the phospholipid composition of the motor cortex with age made this brain area the most stable of the areas studied in terms of its lipid composition.

Interestingly, the cerebellum is often used as a control tissue in studies examining changes in the membrane lipid composition associated with Alzheimer’s disease. Its use is largely due to its relative absence of intra- and extracellular protein aggregates compared to other brain regions [8,9]. The motor cortex, however, remains relatively unchanged in this regard until the very late stages of this disease [10,11,12]. Indeed, few lipid changes have been found to be associated with the pathogenesis of Alzheimer’s disease in the cerebellum compared to other brain regions [13,14,15,16,17], with only white matter showing any significant loss of phospholipids [13]. In the case of phospholipid classes, a previous study found no changes in the overall levels of PC or PE with advancing age [18], a finding which is similar to that found in the present study for all the brain regions examined here, including the cerebellum. However, the present study did find numerous changes in molecular phospholipids with age for all three phospholipid classes (PC, PE and PS). Based on the sheer number of phospholipid changes with age in the cerebellum (plus its substantial cytoarchitectural differences compared to other areas of the brain), an alternative control for studies on the lipid compositional changes in developing Alzheimer’s might be the motor cortex, with its more similar structure to other brain areas and its very stable phospholipid composition at the molecular level during ageing.

The changes in the fatty acid composition of phospholipids in the cerebellum with age were largely driven by increases in the omega-3 (or n-3) molecule DHA, a fatty acid well-known for its abundance in brain tissue (see [2] for a review of the fatty acid composition of various tissues, including the brain) and for being essential to brain function, despite being the fatty acid most prone to oxidation [19]. A prior study examining mitochondrial membranes found that 80% of the peroxidative capacity of the human brain resided in a single-tissue phospholipid, i.e., PE 18:0_22:6 [20]. This same molecule made up 29% of the PE molecules in the microsomal and mitochondrial membranes of the cerebellum and increased by 20% from age 20 to 100 years. Similarly, another major molecule, PS 18:0_22:6 (at 42% of the PS), increased by 49% during ageing. Not surprisingly, an analysis of these changes in the cerebellum showed a net increase in DHA for both the PE and PS classes of both membrane fractions. The increase in DHA in the mitochondrial fraction occurred at a significantly faster rate than that of the microsomal fraction, mainly due to PE molecules containing DHA that increased at a rate of 0.09% per year, versus 0.06% per year in the microsomal membrane (the DHA within the PS molecules increased at similar rates in both membrane fractions). PC was the only phospholipid class that did not show an increase in DHA, but this class is well-known for its low level of DHA-containing molecules [20].

The cerebellum also displayed a fatty acid profile where phospholipids that contained omega-6 (or n-6) polyunsaturated fatty acids, namely arachidonic acid (AA; 20:4) and adrenic acid (22:4), tended to decrease with age. Significant losses of adrenic acid were seen in all phospholipid classes for both membrane fractions, with the single exception of the PC in the microsomal membrane. Decreases in adrenic acid and AA were present in both the PE and PS molecules in all membrane fractions (*p* = 0.041–0.00002). Decreases in n-6 (adrenic and AA) and increases in n-3 polyunsaturates (namely DHA) were also observed in the other brain areas previously examined as part of the overall study. The prefrontal cortex demonstrated decreases in AA for both the mitochondrial and microsomal membranes [3], whereas the hippocampus [4] showed a decrease in AA for the mitochondrial membrane. The entorhinal cortex [5] showed a similar trend, but not at the level of significance. Among the different brain areas studied, the largest age-related increase in phospholipid molecules was for PS 18:0_22:6 in the cerebellum, prefrontal cortex and hippocampus (mitochondrial membrane only), followed by PE 18:0_22:6 in the cerebellum alone. The largest decreases with age were seen for PS 18:0_18:1 within the cerebellum, PE 18:0_20:4 in the cerebellum and prefrontal cortex, and PS 18:0_22:4 in the cerebellum and hippocampus. The only phospholipid molecule to show a significant age-related change across all five brain areas studied was PC 16:0_18:2, which increased with age.

As previously noted, this set of studies found that the phospholipids in some brain areas showed a relative loss of AA and adrenic acid and a gain in DHA with age. In the cerebellum, entorhinal cortex, hippocampus and prefrontal cortex, the molecules that accounted for most of the changes with age primarily stemmed from the PS and PE molecules. PS molecules accounted for 67% and 42% of the significant changes in the microsomal and mitochondrial membrane fractions, respectively, and PE accounted for 33% of the changes in both membrane fractions. The area most resistant to changes in the relative compositions of AA, adrenic acid and DHA was the motor cortex, with no significant decreases in either AA or adrenic acid, and no significant increase in DHA, with the exception of a slight decrease in AA in PC (*p* = 0.043) and PE (*p* = 0.030) molecules in the microsomal membrane. The entorhinal cortex was also quite resistant to these changes, with the only significant decrease being adrenic acid in the PC of mitochondrial membrane and PS of microsomal membrane. These findings of distinct age-related changes in the major polyunsaturated fatty acids associated with phospholipids could be interpreted in two ways: firstly, phospholipid fatty acid composition changes with age, or secondly, individuals with phospholipids that contain less AA and/or adrenic acid and increased levels of DHA in some areas of the brain (e.g., cerebellum and prefrontal cortex) live longer.

Some of the changes in the phospholipids in areas of the human brain that were found to occur with age are likely to impinge on brain function. For example, phospholipids with saturated and monounsaturated fatty acids have been shown to reduce the peroxidation of polyunsaturated fatty acids associated with membranes by extending the lag phase leading into peroxidation [21]. Certainly, the increase with age in the major PC molecule, PC 16:0_18:1, in the mitochondrial membranes (where free radicals are generated) of the prefrontal cortex, entorhinal cortex and hippocampus (but lacking in the microsomal membranes) would support this idea. This same action would also be facilitated by an increase in the major saturated phospholipid PC 16:0_16:0 in the microsomal membrane of the cerebellum. Another phospholipid of interest is PS 18:0_18:1, which acts as a membrane stabiliser for the Na/K-ATPase active transporter [22]. This PS molecule decreased in both membrane fractions of the cerebellum. An increase in the activity of the Na/K-ATPase, a major energy consumer of the brain, has also been linked to phospholipids that contain DHA, such as PE 16:0_22:6 and PS 18:0_22:6 [22], which showed an increase with age (either one or the other) in both membrane fractions of all brain areas except for the motor cortex.

The changes in the acyl composition (increases in DHA and decreases in AA and adrenic acid) in the various areas of the brain examined can be used to determine what form of ageing might occur in the brain. Currently, there are two main ageing theories: first, the mitochondrial free radical theory of ageing [23], and second, the inflammageing theory [24]. The free radical theory of ageing would suggest a loss of polyunsaturates in membranes (particularly mitochondria) with age over time due to the highly oxidative nature of these fatty acids [19]. The inflammageing theory would suggest that n-6 polyunstaurates (notably AA) in membranes are reduced in their role as precursors for generating proinflammatory molecules (prostaglandins, leukotrienes and thromboxanes). Adrenic acid might also be expected to decrease in undergoing β-oxidation within peroxisomes to form AA, whereas DHA, with its low affinity for phospholipase A2 and COX-1 and -2 compared to AA, might be less impacted. The fact that DHA increased and AA and adrenic acid decreased in many areas of the brain would suggest that chronic low-grade inflammation might be a driving force behind normal ageing within the human brain. The lack of change in some brain areas also suggests that these areas (e.g., motor cortex) are more resistant to these changes than others.

Most of the changes found in the present set of studies tend to agree with previously published results examining comparable areas, including a recent study of the human frontal cortex [25]. All studies tend to agree that the mole percentage of phospholipid classes in the human brain are conserved during ageing (studies include the frontal cortex, cerebellum and hippocampus [18,26,27,28]). Some previous studies have also noted some decreases in AA and adrenic acid, plus an increase in DHA, with age in the human brain [27,29,30,31,32]. However, differences exist; notably, the decrease found in AA in both membrane fractions of the prefrontal cortex do not agree with the results of some other studies [27,29,30]. This disagreement may be associated with the different areas of the frontal cortex that were examined. The present study examined the dorsolateral prefrontal cortex (BA 9/42) and primary motor cortex (BA 4), whereas other studies have examined the frontal eye field (BA 8) [27]), the premotor cortex (BA 6) [32], the orbitofrontal cortex (BA 10) [31] or undisclosed sections of the frontal cortex [29,30], with each area having unique differences that could influence its phospholipid composition in response to age.

As noted in the methods, some donors over the age of 60 were observed at Braak stage I or II (of the six progressive stages of Braak). Lower levels of the Braak stage are most likely to affect the entorhinal region of the brain. Regarding any influence of the Braak stage on the phospholipid composition of the entorhinal cortex, it is notable that, along with the motor cortex, this area of the brain was one of the least affected by age. It is also worth noting that the subjects used in this study died primarily from cardiac (72%) and respiratory (16%) diseases and other causes (including trauma, cancer, embolism and infection), with no deaths attributed to any neurological issues. The present analysis used a basic fractionation method from frozen samples that could allow for some contamination between membrane fractions. One of these could be the mitochondrial fraction having some endoplasmic reticulum (ER) from the mitochondria-associated region where phospholipid metabolism occurs [33]. It needs to be kept in mind that the microsomal membrane is an artificial construct that includes many different membrane types (ER, Golgi, plasma and vesicular membrane systems). However, even with these caveats, there were still differences between the two membrane fractions as indicated by the heat map analysis, suggesting that changes with age in the phospholipid composition of the mitochondrial membrane might be slightly different to those of other membranes.

## 5. Conclusions

This work found that the overall percentage of PC, PE and PS in all brain areas (with a minor exception in the entorhinal mitochondrial membrane) did not change with age. However, the acyl composition of phospholipids showed considerable change. The cerebellum was unique among the different areas of the brain studied in having many phospholipid molecules, including several major phospholipids, that underwent increases and decreases with age (e.g., in both membranes, increases were observed in PE 18:0_22:6, PE 16:0_22:6 and PS 18:0_22:6 and decreases were observed in PE 18:0_20:4, PE 18:0_22:4, PS 18:0_18:1 and PS 18:0_22:4; in the microsomal membrane only, an increase was observed in PC 16:0_16:0). In contrast, the motor cortex had less than 40% of its phospholipids undergoing changes with age, and none of those were major phospholipids. Likewise, the entorhinal cortex was found to be an area of the human brain that appears to be resistant to major changes in its phospholipid composition with age. In the cerebellum and prefrontal cortex, the microsomal and mitochondrial membranes demonstrated reductions in their level of AA and adrenic acid levels and increases in their DHA levels. Even those brain areas that were more resistant to change showed some significant shifts, with the hippocampus having less AA and the entorhinal cortex having less adrenic acid in their mitochondrial membranes with age.

## Figures and Tables

**Figure 1 nutrients-14-02495-f001:**
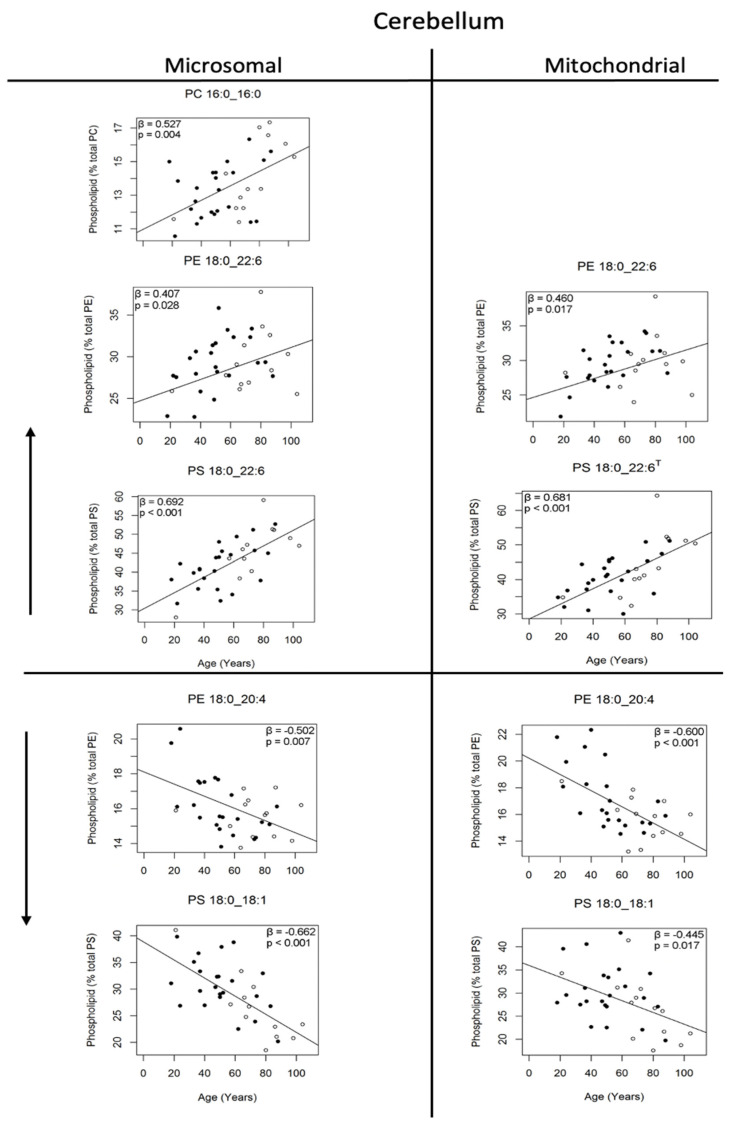
Phospholipids of the cerebellum, representing greater than 10% of each phospholipid class, that changed significantly in the microsomal and mitochondrial membranes with age; ^T^ indicates dependent variables transformed for linear regression. Regression models were adjusted for sex: males (●) and females (○). The beta coefficient and *p* value are provided on each plot (*n* = 33–36).

**Figure 2 nutrients-14-02495-f002:**
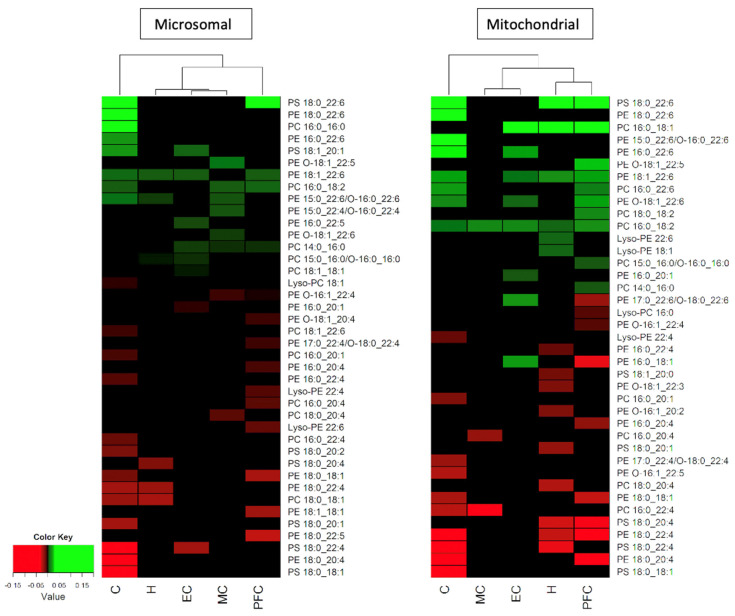
A summary of the significant changes seen with normal ageing in the phospholipid composition (% of the phospholipid class) in the microsomal and mitochondrial membranes of the cerebellum (C), motor cortex (MC), entorhinal cortex (EC), hippocampus (H) and prefrontal cortex (PFC). Increases in phospholipids with age are shown in green, and decreases in red. The hierarchical clustering of the five brain regions is shown by the dendrogram at the top of each heat map.

**Table 1 nutrients-14-02495-t001:** The relative abundance (% of the PC, PE or PS) and percentage of change (% change) of the membrane phospholipids that changed in the cerebellum and motor cortex between the ages of 20 and 100 years.

		Fatty Acid Combinations	Microsomal				Mitochondrial			
			Cerebellum		Motor Cortex		Cerebellum		Motor Cortex	
			% of the PC, PE or PS	% Change	% of the PC, PE or PS	% Change	% of the PC, PE or PS	% Change	% of the PC, PE or PS	% Change
Increase	PC	14:0_16:0	-	-	1.42 ± 0.02	15%	-		-	
		16:0_16:0	13.55 ± 0.31	30%	-		-		-	
		16:0_18:2	1.18 ± 0.05	54%	1.25 ± 0.05	51%	1.10 ± 0.05	39%	1.26 ± 0.05	59%
		16:0_22:6			-		4.00 ± 0.03	20%	-	
	PE	16:0_22:6	6.21 ± 0.11	25%			6.27 ± 0.12	34%	-	
		18:0_22:6	29.27 ± 0.56	20%			29.54 ± 0.56	21%	-	
		18:1_22:6	2.86 ± 0.05	26%			2.95 ± 0.07	33%	-	
		15:0_22:4/ O-16:0_22:4	-		0.65 ± 0.07	86%			-	
		15:0_22:6/ O-16:0_22:6	1.66 ± 0.09	61%	0.73 ± 0.06	83%	1.73 ± 0.12	263%	-	
		O-18:1_22:5	-		1.36 ± 0.11	75%			-	
		O-18:1_22:6	-		1.99 ± 0.04	16%	2.74 ± 0.06	22%	-	
	PS	18:0_22:6	42.91 ± 6.67	48%			42.16 ± 1.22	50%	-	
		18:1_20:1	1.89 ± 0.70	129%					-	
Decrease	PC	Lyso-18:1	0.86 ± 0.05	10%					-	
		16:0_20:1	1.54 ± 0.22	23%			1.48 ± 0.21	27%		
		16:0_20:4	-				-		3.59 ± 0.10	17%
		16:0_22:4	2.50 ± 0.07	22%			2.89 ± 0.23	25%	1.95 ± 0.15	41%
		18:0_18:1	7.55 ± 0.17	18%			-		-	
		18:0_20:4	-	-	1.71 ± 0.08	28%	-		-	
		18:1_22:6	1.12 ± 0.03	59%			-		-	
	PE	Lyso 22:4					0.51 ± 0.03	38%	-	
		16:0_22:4	0.78 ± 0.04	38%			-		-	
		18:0_18:1	2.83 ± 0.08	27%			2.79 ± 0.08	28%	-	
		18:0_20:4	15.97 ± 0.26	16%			16.71 ± 0.38	25%	-	
		18:0_22:4	3.49 ± 0.14	36%			3.36 ± 0.13	39%	-	
		O-16:1_22:4	-		0.52 ± 0.03	45%	-		-	
		O-16:1_22:5					1.09 ± 0.07	54%	-	
		17:0_22:4/ O-18:0_22:4					1.31 ± 0.07	40%	-	
	PS	18:0_18:1	29.88 ± 6.85	38%	-		28.84 ± 1.06	30%	-	
		18:0_20:1	2.13 ± 1.13	56%	-		-		-	
		18:0_20:2	1.10 ± 0.65	38%	-		-		-	
		18:0_22:4	6.12 ± 1.64	28%	-		6.25 ± 0.23	37%	-	

Brains were obtained from subjects aged 18 to 104 years, but the comparative analysis was based on the differences between the ages of 20 and 100 years (adjusted for gender). The % of the PC, PE or PS refers to the average percent abundance of each phospholipid molecule within each phospholipid class. % change refers to the percentage change each phospholipid molecule underwent between ages 20 and 100 years. All values are means across the cohort ± SEM. (-) indicates phospholipid not present in that particular membrane fraction.

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
