# Peer review of "Changes in Phospholipid Composition of the Human Cerebellum and Motor Cortex during Normal Ageing"

_nutrients, 2022, doi:10.3390/nu14122495_

Round 1

Reviewer 1 Report

The authors showed changes in phospholipid composition with age in mitochondrial and microsomal membrane of the human cerebellum and motor cortex of subjects aged 18-104 years. With age, there was an increase in PE 18:0_22:6 and PS 18:0_22:6 and in PC 16:0_16:0 only in microsomal membrane and a decrease in PE 18:0_20:4 and PS 18:0_18:1. It is observed that the cerebellum is a unique area in phospholipids undergoing change with age, whereas the motor cortex is highly resistant. 

The manuscript is clear, comprehensive, presented in a well-structured manner and and the manuscript could be of considerable interest to the scientific community. The figures are clear and easily interpretable and understandable. 

In the abstract (line 13) and in introduction (line 57) the authors were reported “18-104 years” of age, but in statistical analysis (line 157) you indicated “20-100 years” and also in table 1. The authors should perhaps indicate the same age or explain this difference.

The manuscript, it is ok, but there are only a few minor revisions:

In Materials and Methods, in “brain tissue”, line 91-93, the authors wrote “Brains from donors over the age 91 of 60 years underwent a modified Braak staging screen with stage I and II being observed 92 in elderly donors”. It is perfect, but they maybe should explain why it is done, and it is important to discuss it in discussion to point out that the changes do not due to pathological dysfunctions.

The abbreviations were not used at various points in the article, it is needed to verify and correct.

In line 355 and 464, to correct “faction” in “fraction”.

Author Response

Responses

1) “18-104 years” of age, but in statistical analysis (line 157) you indicated “20-100 years” -have included more explanation into text of manuscript including:

Methods

 Although brains were obtained from subjects aged from 18 to 104 years, all comparative analysis was based on changes (using linear regression) at 20 years compared to those at 100 years, an 80-year adult age span.

Results

Table 1 summarises all age-related changes in PC, PE and PS in both the microsomal and mitochondrial enriched fractions of the cerebellum and motor cortex between the ages of 20 to 100 years.

Table 1 footnotes

Brains were obtained from subjects aged 18 to 104 years, but comparative analysis was based on differences between the ages of 20-100 years (adjusted for gender). % PC, PE or PS refers to the average percent abundance of each phospholipid molecule within each phospholipid class. * % change refers to the percentage change each phospholipid molecule underwent between ages 20 to 100 years. All values are means across the cohort ± SEM

2) Explain Braak implications – have included more information on Braak staging and implications (Note: Braak screening was performed by the New South Wales Tissue Resource Centre with no other information provided to us with brains deemed neurologically normal for age).

Included in methods - Brains from donors over the age of 60 years underwent a modified Braak staging screen (by New South Wales Tissue Resource Centre) with stage I and II (of the six progressive stages) being observed in some elderly donors but deemed neurologically normal for age.

Included in Discussion - As noted in the methods, some donors over the age of 60 were observed at Braak stage I or II (of the six progressive stages of Braak). Lower levels of Braak stage are most likely to affect the entorhinal region of the brain. Regarding any influence of Braak stage on phospholipid composition of the entorhinal cortex, it is notable that along with the motor cortex this area of the brain was one of the least affected by age. It is also worthwhile noting that subjects used in this study died primarily from cardiac (72%) and respiratory (16%) diseases, then other causes (including trauma, cancer, embolism, and infection) with no deaths attributed to any neurological issues.

3) Abbreviations – have increased use where appropriate except in titles to improve the readability of the paper.

4) error - faction corrected to fraction (thanks)

Reviewer 2 Report

This manuscript reports age-dependent changes of phospholipids in human cerebellum and motor cortex. It is a follow-up work of the authors' previous studies on the changes of phospholipid composition in the aging human brain. The reported findings are a valuable piece for relevant field.

The following comments/suggestions might help to improve the manuscript.

(1) in table 1, the authors give the quantitative information on different phospholipids in the cerebellum and motor cortext. more details/explanations on the data should be provided. e.g., what does the column % of PC or PE or PS mean? are the numbers (e.g., 13.55 +/- 0.31) mean +/- SEM of data from all the subjects? And what does % change mean? difference between age 20 and age 100? 

(2) similar problems also happen in the maintext describing the data in table 1. please revise all relevant sentences.

(3) Figure 2 and the sections describing it (e.g., lines 389 - 440) should be moved from the discussion part to the results part. 

(4) the discussion is too long, and most part of it is simply a re-description of the results. this is not necessary. the authors should make the discussion more concise. in addtion, the discussion shoud focus more on the potential meaning of their findings, e.g., what might be the reasons for the changes of phospholipids during aging, why different brain regions show different chagnes, and what are the relevance of these changes on brain aging? 

Author Response

Responses

1 and 2) what does % change mean etc? – the reviewer is correct it means the percent change from 20 to 100 years – but to increase explanation of these terms we have included more definition in the methods, results and Table 1 footnotes as outlined in comments to Reviewer 1 – we have also changed the title of Table 1 to better reflect what is encompassed ‘Table 1: The relative abundance (%PC, PE or PS) and percentage of change (% Change) of those membrane phospholipids that changed in the cerebellum and motor cortex between the ages of 20 to 100 years.’

3) Figure 2 and the sections describing it (e.g., lines 389 - 440) should be moved from the discussion part to the results part. – have moved this very large section to the results and in the process reduced the discussion as suggested by this reviewer (see comment below).

4) discussion is too long – in line with reviewers wishes we have reduced the discussion (~35%) by moving an appropriate section (as indicated by reviewer) from the discussion to the results.  

Reviewer 3 Report

Comments to Authors of “Changes in Phospholipid Composition of the Human Cerebellum and Motor Cortex during Normal Ageing”

·         Introduction was well written and provided a brief background of the topic.

·         In the methods section 2.2, it is stated that ‘normal post-mortem brain samples were obtained’, however it also states that for donors over 60 years of age, modified Braak stage screening was used with stages 1, 2 observed. As stages 1,2 can detect early changes in lower brainstem in Parkinson’s disease and NFTs in entorhinal cortex in Alzheimer’s disease, it is suggested that some donors may not have been completely free of neurological diseases and thus the possibility of early changes due to such diseases cannot be ruled out. Further clarification of the Braak screening results is needed.

·         In the Results, section 3.2 the Sub-heading should be edited to include the other phospholipids, PE and PS, as they ae also discussed in this section.

·         Overall, the results are well-written and graphically displayed adequately. The changes in lipidomic findings are very interesting and easily understood.

·         The Discussion section should address more functional interpretation of the findings, specifically the conservation of DHA containing phospholipids in cerebellum and other areas, and likewise the loss of ARA and adrenic acid. How does such conservation apply functionally to aging adults? E.g. It is well known that phospholipids with DHA are significantly decreased in Alzheimer’s disease in areas of hippocampus, entorhinal cortex and others…

What could it mean functionally to observe a loss of ARA in aging based on previous studies of ARA’s role in brain function?

·         The Conclusion should also provide a brief functional interpretation of the findings. The last sentence of the Conclusion is unclear and should be a concise and impactful statement for the reader.

·         Minor comment: typo online 355 ‘faction’ should be fraction.

Author Response

Responses

1) Further clarification of the Braak screening results is needed – this has been improved with inclusion of an extended section in methods and section on implication in discussion as outlined in responses to Reviewer 1

2) Section 3.2 the Sub-heading should be edited to include the other phospholipids, PE and PS – The original opening section of section 3.2 discussed the most abundant phospholipid molecules in each class i.e. PC, PE and PS and this has produced the mistaken impression that Section 3.2 will discuss changes in all 3 phospholipid classes in detail, but it doesn’t. To alleviate the problem a new 3.2 section has been added entitled ‘ Abundant Phospholipid Molecules of the Human Cerebellum and Motor Cortex’ that takes the front end of the old Section 3.2 that discusses the most abundant phospholipid types in each of the two brain areas, then the remaining portion of section 3.2  has been made into section 3.3 that discusses changes in PC molecules only as indicated by the heading title and section 3.4 discusses changes in both PE and PS molecules as indicated by the section title. So its now much clearer (hopefully)

3 and 4) More functional interpretation of the findings in discussion (notably increased DHA and decreased AA) in association with implications for ageing. What could it mean functionally to observe a loss of ARA in aging  – within the constraints of what is known we have included a paragraph on some of the functional implications of the changes in the polyunsaturates associated with age, specifically the two main ideas on ageing i.e. the free radical oxidative theory and the inflammageing therory of ageing where the changes seen in the different brain areas at this point would tend to support the later theory.

5) The last sentence of the Conclusion is unclear – agree – have removed it as it added nothing new

6) typo online 355 ‘faction’ should be fraction – done (thanks)